# Autonomous Driving of Mobile Robots in Dynamic Environments Based on Deep Deterministic Policy Gradient: Reward Shaping and Hindsight Experience Replay

Minjae Park [1], Chaneun Park [2] and Nam Kyu Kwon [1,*]

1 Department of Electronic Engineering, Yeungnam University, Gyeongsan 38541, Republic of Korea; alswo6700@yu.ac.kr
2 School of Electronics Engineering, Kyungpook National University, Daegu 41566, Republic of Korea; chaneun@knu.ac.kr
* Correspondence: namkyu@yu.ac.kr; Tel.: +82-53-810-3095

**Abstract:** In this paper, we propose a reinforcement learning-based end-to-end learning method for the autonomous driving of a mobile robot in a dynamic environment with obstacles. Applying two additional techniques for reinforcement learning simultaneously helps the mobile robot in finding an optimal policy to reach the destination without collisions. First, the multifunctional reward-shaping technique guides the agent toward the goal by utilizing information about the destination and obstacles. Next, employing the hindsight experience replay technique to address the experience imbalance caused by the sparse reward problem assists the agent in finding the optimal policy. We validated the proposed technique in both simulation and real-world environments. To assess the effectiveness of the proposed method, we compared experiments for five different cases.

**Keywords:** deep deterministic policy gradient; multifunctional reward shaping; hindsight experience replay; mobile robot; autonomous driving

## 1. Introduction

Mobile robots have recently played a key role in the field of autonomous driving [1–5], and in application fields such as delivery and service operations in buildings, hospitals, and restaurants [6–8]. To fulfill these roles, three essential processes are required: perception of the surrounding environment, a decision-making process to generate a path to the destination, and a control process for the motion of the robot. First, in the perception process, the robot obtains information about its surroundings using sensors, such as cameras or LiDAR. Next, the decision-making process involves a path-planning procedure. One of the representative techniques is Simultaneous Localization and Mapping (SLAM), allowing the robot to navigate based on the surrounding map [9]. Finally, in the control process, SLAM regulates its motions by following the reference path generated in the decision-making process. Many studies have been conducted on SLAM-based autonomous driving [10]; however, there is a limitation in the complexity of implementation due to the necessity of multiple sensors or deep learning for accurate surrounding perception and navigation in environments with unexpected dynamic obstacles. To alleviate the issue associated with dynamic obstacles, Dang et al. [11] modified the SLAM by implementing sensor fusion and dynamic object removal methods. They achieved accurate position estimation and map construction through integrated sensor-based dynamic object detection and removal techniques, including radar, cameras, LiDAR, and more. However, if an error occurs in the process of estimating the position and motion of the dynamic object, it may be difficult to remove the object accurately. In addition, the complexity of implementation due to the necessity of synchronizing multiple sensors remains, making it challenging to apply even when improving dynamic environments. Xiao et al. [12] proposed the Dynamic-SLAM

technology, which combines SLAM and deep learning. The authors of [12] proposed a method using a single-slot detector based on a convolutional neural network (CNN) to detect dynamic objects. They enhanced the detection recall through a compensation algorithm for missed objects. As a result, this method demonstrated an improved performance in the presence of dynamic obstacles based on a visual SLAM. However, this approach still has limitations in that an environment map must be drawn and CNN requires a large amount of data. To address these issues, many researchers have studied the use of reinforcement learning for autonomous driving. Unlike SLAM technology, which requires additional adjustments to handle dynamic obstacles, reinforcement learning-based autonomous driving has the advantage of adapting to changes in the environment by utilizing the optimal policy. Applying reinforcement learning allows finding the optimal policy needed for autonomous driving with just a single sensor. Even when considering the additional burden associated with deep learning, it can reduce the tasks required for data collection. Therefore, it can be considered less complex to implement than SLAM. These advantages have attracted the interest of many researchers [13–20].

Reinforcement learning is akin to mimicking the direct engagement and experiential type of learning found in humans. Generally, humans employ two types of learning methods: indirect learning through observation and direct learning through hands-on experience. Traditionally, machine learning, which simulates the indirect learning method, has produced outstanding results in object recognition and image classification. Similarly, just as people develop the ability to make split-second decisions through experience, research is ongoing to develop neural networks capable of quick decision-making through reinforcement learning, aimed at handling complex tasks. The main goal of reinforcement learning is to find an optimal policy that achieves an objective through the interaction of an agent with the environment. The agent observes the environment and determines the optimal action. After executing the action, the agent receives a reward as feedback. Based on these processes, the agent finds the optimal policy that guarantees the maximal cumulative reward. Generally, the rewards obtained during experience collection can only provide meaningful information after the episode has ended. Therefore, in this case, finding the optimal policy can be difficult because most rewards are meaningless. This problem is referred to as the sparse reward problem and becomes more pronounced in the autonomous driving of mobile robots in environments with dynamic obstacles. It is challenging to simultaneously achieve the goals of reaching the destination and avoiding obstacles, as these goals are included. Two techniques can be applied to alleviate this problem. The first approach, reward shaping, is a method of complementing the reward system with a specific one that can provide sufficient information about the goals based on domain knowledge of the task. Jesus et al. [21] successfully implemented reward shaping for the indoor autonomous driving of mobile robots. However, there was a limitation of not adequately addressing the goal of obstacle avoidance by reflecting only information about the destination in the reward function. The second technique is the hindsight experience replay (HER) method, which generates alternate success episodes by extracting partial trajectories from failed episodes [22]. The HER can increase the number of successful experiences in the learning database by reevaluating failed experiences as alternate success experiences with virtual objectives. Consequently, it promotes the exploration of various routes, increasing the probability of reaching the actual destination. In our previous study, we employed the HER to implement the autonomous driving of a mobile robot based on reinforcement learning. The agent was trained in a simple driving environment within the simulation. We demonstrated that the proposed method operates effectively in both simulated and actual environments [23]. The HER has also been widely applied in the fields of mobile robotics and robot arm control [24–28]. Both reward shaping and the HER have individually been used to implement reinforcement learning-based autonomous driving schemes in dynamic environments. However, to the best of our knowledge, no attempts have been made to utilize both methods for handling dynamic environments, which was the objective of the present study.

We designed a reinforcement learning method for dynamic environments with moving obstacles by considering the concepts of both multifunctional reward shaping and the HER. First, by adopting the concept of reward shaping, specific information about the environment is reflected in the reward function to guide the agent toward goals. For navigation to the destination, the reward function was designed based on destination information, and for obstacle avoidance, the reward function was designed based on obstacle information. Additionally, we employed the HER, which involves the re-generation of successful episodes, addressing the data imbalance issue between successful and failed episodes. Consequently, the proposed method can improve the policy optimization process. To validate the effectiveness of the proposed method, we performed an autonomous driving experiment to compare the following methods based on the deep deterministic policy gradient (DDPG):

(1) Only DDPG [29].
(2) Only reward shaping (goal-based) [21].
(3) Only reward shaping (proposed method).
(4) Only HER [23].
(5) Proposed method.

## 2. Preliminaries

### 2.1. Deep Deterministic Policy Gradient

The DDPG is a reinforcement learning algorithm that can handle continuous action spaces, with a deterministic policy μ used to determine agent behavior [30]. Most policy gradient algorithms used to handle continuous action spaces utilize the policy directly instead of using a value function. In this case, the stochastic policy $\pi$ is used to determine the agent's behavior using a probability distribution. However, the basic algorithm of DDPG, the deterministic policy gradient (DPG) algorithm, demonstrated that the policy gradient method can be used even when μ is used instead of $\pi$ [31]. The DPG not only offers the advantage of increasing convergence in the learning process to find the optimal action policy but also plays an effective role in the continuous action space. The DDPG algorithm employs two prominent techniques of the deep Q-network (DQN) to increase the efficiency and stability of learning based on the actor-critic structure of the DPG [32]. The first prominent technique of DQN is the experience replay, which stores experience data transitions in memory for reuse. This technique increases learning efficiency owing to data reuse and mitigates adherence to suboptimal policies by reducing the correlation between data owing to the random selection of training data. The second technique is a target network separation that generates target networks with the same structure as the critic and actor networks. This technique enables the stable updating of network parameters by reducing the instability of reclusively generated target values. Each set of target network parameters—including the critic network parameter $\theta_Q$ and target critic network parameter $\theta_\mu^-$, as well as the actor network parameter $\theta_\mu$ and target actor network parameter $\theta_Q^-$—is updated using the soft update method with a ratio parameter $\tau$:

$$\theta_Q^- \leftarrow \tau\theta_Q + (1-\tau)\theta_Q^-, \tag{1}$$

$$\theta_\mu^- \leftarrow \tau\theta_\mu + (1-\tau)\theta_\mu^-. \tag{2}$$

In the process of determining the action, the DDPG accounts for Ornstein–Uhlenbeck noise [33], and $\mathcal{N}_t$ in the output of the actor network:

$$a_t = \mu(s_t) + \mathcal{N}_t. \tag{3}$$

Lei et al. [29] implemented the DDPG algorithm for mapless navigation for a mobile robot in both simulations and the real world. It demonstrates satisfactory performance in reaching the intended destination compared to traditional map-based navigation. However, the considerable number of episodes needed to find the optimal policy raises a potential

cost concern. Therefore, this study applied a technique to quickly find a policy that can adapt to dynamic environments by simultaneously employing the HER and reward shaping methods.

### 2.2. Hindsight Experience Replay

The HER is a technique that generates positive experiences by re-evaluating failed episodes. Generally, as an agent collects training data through exploration, most of these data correspond to failed episodes. It may be challenging to optimize a policy that fits the objective. The HER can be used to overcome these difficulties. After an episode ends, parts of the trajectories of failed episodes are extracted and converted into success trajectories. In this process, a specific state is selected from the extracted trajectory as the new virtual destination. Then, starting from the initial state of the extracted trajectory to the new destination, the reward for the new successful trajectory is recalculated and stored in memory as if the goal had been achieved. As successful trajectories are added to the database, it has the effect of exploring new spaces. These experiences promote actual exploration, increasing the likelihood of reaching the real destination and ultimately improving the probability of finding the optimal policy.

The technology applied in this study was previously introduced in the authors' earlier work [23]. In [23], a method integrating the HER technique to assist in finding the optimal policy was proposed and demonstrated its effectiveness in both simulation and real-world environments without obstacles. However, we recognize the limitations in handling dynamic obstacles and further extend the previously applied HER technique by integrating reward shaping. The proposed method in this study not only helps in finding the optimal policy but also adapts to dynamic environments.

### 2.3. Reward Shaping

Reward shaping is a technique used to design or adjust a reward system for reinforcement learning. The reward system, which provides feedback according to the action executed by the agent, can be either sparse or dense according to the specificity of the reward. The sparse reward system does not provide specific information regarding the goals in the process of finding an optimal action policy. For example, the sparse reward system in a chess game offers a reward of 0 for each individual move, and only the final reward provides positive or negative information based on the game outcome. In this case, it may be necessary to add a specific reward function that accounts for the values of individual pieces. In this way, domain knowledge about the task is required to induce the intended action in reward shaping. The specific update of the reward system based on domain knowledge is referred to as a dense reward system. In such a reward system, the rewards are explicitly specified for actions taken by the agent at each state, providing sufficient information about the goals. This process is defined by Equation (4), where $r'$ is an updated reward system using reward shaping, r is the original reward system, and $f(s, a)$ is a reward function based on the state and action of the reward-shaping technique.

$$r' = r + f(s, a) \tag{4}$$

There is a study that applied reward shaping to the autonomous driving of mobile robots based on reinforcement learning [21]. Jesus et al. [21] proposed a reward-shaping technique based on destination distance to complement the reward system. The proposed approach was validated in experimental environments divided into three stages: stage 1 (obstacle-free environment), stage 2 (environment with fixed obstacles), and stage 3 (environment with additional walls and moving obstacles). The proposed approach effectively operated in stage 1; however, in stage 2 and stage 3, the presence of additional obstacles led to an increase in the required learning episodes to find the optimal policy or failure to find the optimal policy. Therefore, in environments with many dynamic obstacles like the real world, it is anticipated that the efficiency of the proposed technique may be compromised. In this study, recognizing the necessity of obstacle handling, we

propose a method that simultaneously applies the HER and reward shaping by adding obstacle-related reward functions.

## 3. Materials and Methods

### 3.1. Mobile Robot and Environmental Configuration

In this study, we used the TurtleBot 3 Burger, as shown in Figure 1a. The robot is equipped with two Dynamixel motors on the left and right sides, which transfer power to the two wheels. The OpenCR controller is used to control these wheels. Additionally, a laser distance sensor is mounted at the top of the robot, allowing it to measure distances around the robot in a 360° range. The detection distance range of this sensor is 0.12 m to 3.5 m. The system is controlled using a Raspberry Pi 3b+ board.

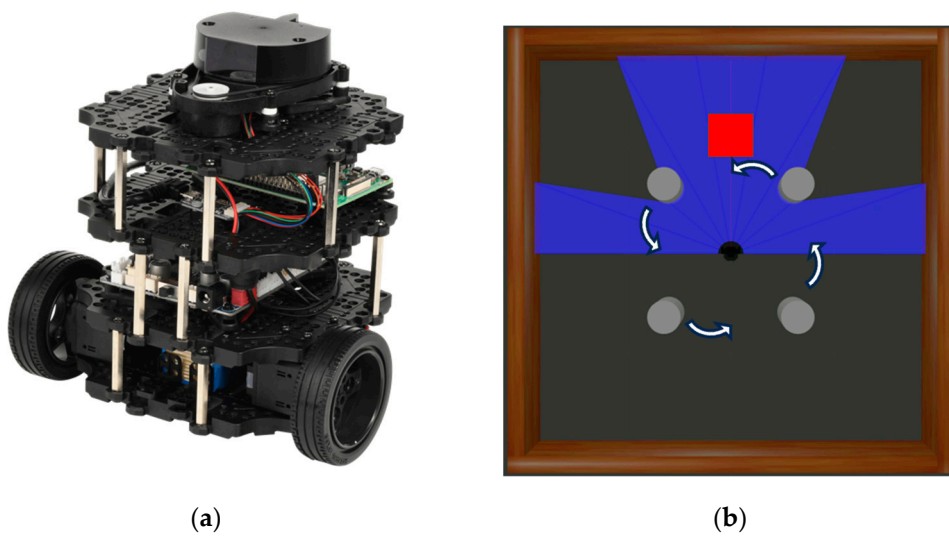

|  |  |
|:--:|:--:|
| (**a**) | (**b**) |

**Figure 1.** (**a**) Turtlebot3 in real-world environment and (**b**) Turtlebot3 and experimental environment with 4 dynamic obstacles in simulation. The red box represens the destination, and the blue area represent the scope of the LDS.

ROS (Robot Operating System) is a software platform for developing robot applications and serves as a meta-operating system used in traditional operating systems such as Linux, Windows, and Android. Communication in ROS is generally categorized into three types: topics, services, and actions. Specifically, topic communication involves one-way message transmission, service communication entails a bidirectional message request and response, and action communication employs a bidirectional message feedback mechanism.

Figure 1b illustrates the Turtlebot3 and the experimental environment within the 3D simulator Gazebo. In this environment, the destination is randomly set when the driving starts. The starting point of the driving is the center of the space, except when reaching the destination, where the navigation restarts from that point. The dynamic obstacles consist of 4 cylindrical structures that rotate with a fixed radius. The Gazebo allows the creation of environments similar to the real world, reducing time and cost in development and enhancing convenience. Moreover, it has good compatibility with ROS.

In this study, as shown in Figure 2, a reinforcement learning system was implemented in the Gazebo simulation using the ROS, utilizing sensor values of the mobile robot and topic communication between nodes. A step is defined as the process in which the robot executes the action determined by the reinforcement learning algorithm, receives a reward, and completes the transition to the next state. As a result of this process, a single transition $(s_t, a_t, r_{t+1}, s_{t+1})$ is generated, consisting of the current state $s_t$, action $a_t$, reward $r_{t+1}$, and the next state $s_{t+1}$. An episode is defined as the trajectory observed when driving begins until the goal is achieved or when failure (collision or timeout) is observed during this process. Success is defined as reaching the destination, while failure includes collisions with obstacles and not reaching the destination within a limited number of actions (timeout). A

trajectory is defined as a connected form of transition resulting from the steps performed within an episode.

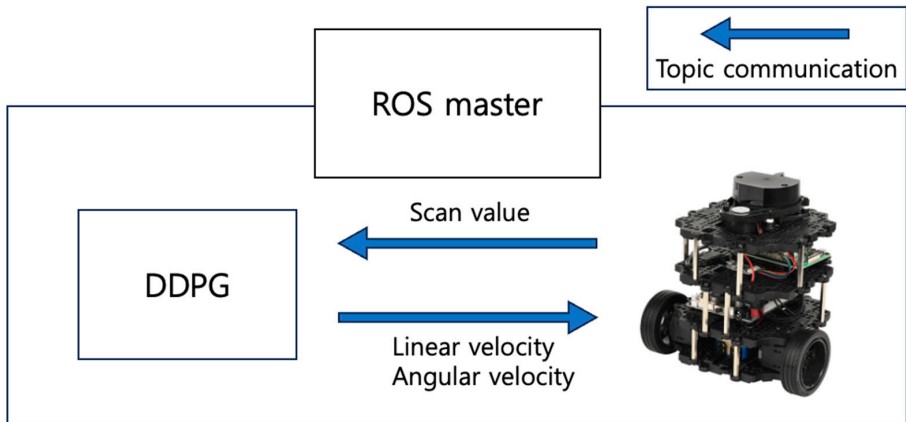

**Figure 2.** Reinforcement learning system using a communication topic communication between the DDPG node and Turtlebot3 node based on ROS.

### 3.2. Reinforcement Learning Parameters

For reinforcement learning, it is necessary to define the state, action, and rewards. The states and actions are described in this subsection, and the reward system is described in detail in Section 3.3. Figure 3 illustrates the states used in the experiments.

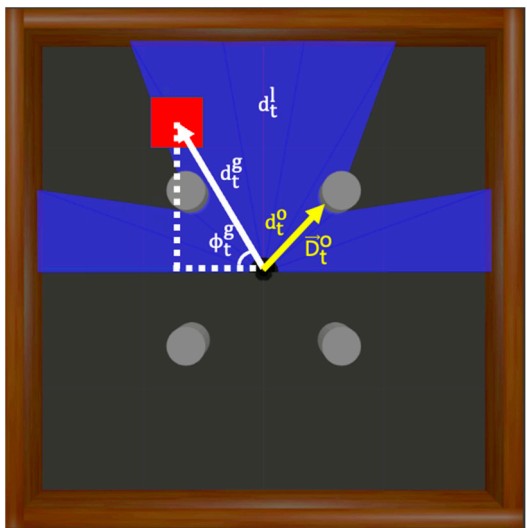

**Figure 3.** Elements constituting the state in the experimental environment include distance information measured by LDS, distance and direction to the destination, past action, and distance and direction to the nearest obstacle.

The state $s_t \in R^{16}$ can be expressed by Equation (5), where $d_t^l \in R^{10}$ is a value measured in front of the robot using a laser distance sensor (LDS) every 18 degrees, over a total of 180 degrees:

$$s_t = \left( d_t^l, d_t^g, \phi_t^g, a_{t-1}, d_t^o, \vec{D}_t^o \right). \tag{5}$$

The variable $d_t^g$ denotes the linear distance between the robot's current coordinates $P = (P_x, P_y)$ and the destination coordinates $P^g = \left( P_x^g, P_y^g \right)$ as follows:

$$d_t^g = \sqrt{\left( P_x^g - P_x \right)^2 + \left( P_y^g - P_y \right)^2}. \tag{6}$$

The variable $\phi_t^g$ denotes the angular difference between the robot's yaw value $\phi_{yaw}$ and the destination:

$$\phi_t^g = \tan^{-1}\frac{\left(P_y^g - P_y\right)}{\left(P_x^g - P_x\right)} - \phi_{yaw}. \tag{7}$$

The variable $a_{t-1} \in R^2$ denotes the immediate previous action, defined as follows:

$$a_{t-1} = (v_{t-1},\ \omega_{t-1}). \tag{8}$$

where $v_{t-1}$ and $\omega_{t-1}$ denote the robot's linear and angular velocities for that action, respectively. The variable $d_t^o$ denotes the linear distance between the coordinates of the robot $(P_x,\ P_y)$ and those of the closest obstacle $P^o = (P_x^o,\ P_y^o)$:

$$d_t^o = \sqrt{(P_x^o - P_x)^2 + \left(P_y^o - P_y\right)^2}. \tag{9}$$

The variable $\overrightarrow{D_t^o}$ denotes the direction of the obstacle closest to the robot:

$$\overrightarrow{D_t^o} = \underset{\theta}{\mathrm{argmin}}\left(d_t^l\right). \tag{10}$$

Figure 4 demonstrates the components of action, and the action $a_t \in R^2$ is constructed from $v_t$ and $\omega_t$, along with noise $\mathcal{N}_t$:

$$a_t = (v_t,\ \omega_t) + \mathcal{N}_t. \tag{11}$$

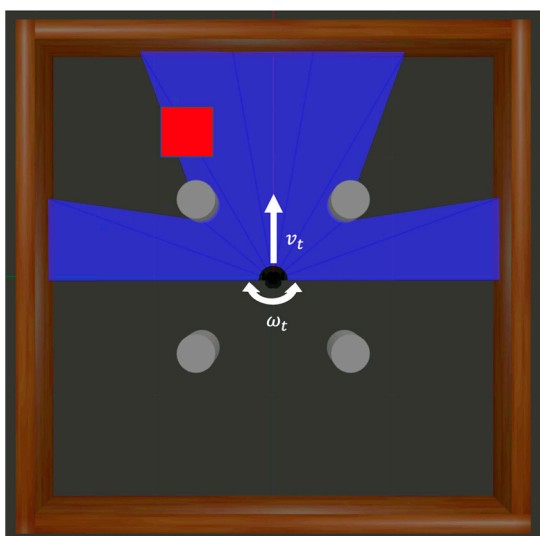

**Figure 4.** Elements constituting the action in the experimental environment include linear and angular velocities of robot.

### 3.3. Design of Reward System

A sparse reward system can be expressed as in Equation (12). If the distance between the robot and its destination is less than 0.15 m after performing an action, it is defined as a success, and a reward value of +500 is returned. On the other hand, if the robot collides with a wall or obstacle, a reward value of $-550$ is returned instead. In cases where no special state transitions occur, as above, all other state transitions following an action receive a reward of $-1$.

$$R_t^{sparse}(s_t,\ a_t) = \begin{cases} 500, & \text{if } d_t^g < 0.15\,\mathrm{m} \\ -550, & \text{if } d_t^o < 0.135\,\mathrm{m}. \\ -1, & \text{otherwise} \end{cases} \tag{12}$$

The proposed reward system expressed in Equation (13) is augmented with reward shaping that includes specific information related to the destination and obstacles. To provide detailed information about the destination, an additional reward is introduced based on the change in distance to the destination. When the distance decreases, a positive reward proportional to the change in distance is generated, whereas when the distance increases, a fixed negative reward is applied instead.

$$R_t^g(s_t, a_t, P^g) = \begin{cases} \alpha_t & \text{if } \left(d_{t-1}^g - d_t^g\right) > 0 \\ -8 & \text{otherwise} \end{cases}, \tag{13}$$

where $\alpha_t = 200\left(d_{t-1}^g - d_t^g\right)$. Distance information associated with obstacles is also included in the reward, ensuring that an optimal policy would avoid moving obstacles:

$$R_t^o(s_t, a_t, P^o) = \begin{cases} \beta_t & \text{if } \left(d_{t-1}^o - d_t^o\right) > 0 \\ -\beta_t & \text{otherwise} \end{cases}, \tag{14}$$

where $\beta_t = 550 \exp[-70(d_t^o - 0.2)]$. By introducing reward shaping, the final reward system ensures that the rewards for reaching the destination and collisions remain the same as in Equation (13), whereas those for other individual actions are expressed by the sum of the following reward functions:

$$R_t^{\text{Dense}}(s_t, a_t) = \begin{cases} 500, & \text{if } d_t^g < 0.15 \text{ m} \\ -550, & \text{if } d_t^o < 0.135 \text{ m.} \\ R_t^g + R_t^o, & \text{otherwise} \end{cases} \tag{15}$$

As shown in Figure 5a, it can be observed that negative rewards increase rapidly when the distance from the current state to the obstacle becomes closer than in the past state. In contrast, as shown in Figure 5b, it can be observed that positive rewards increase rapidly when the distance from the current state to the obstacle becomes farther than in the past state.

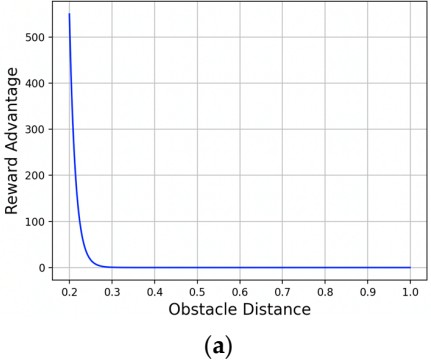
(a)

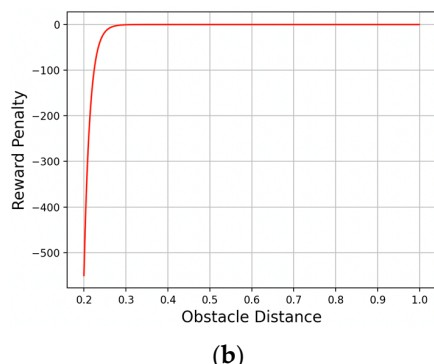
(b)

**Figure 5.** (**a**) A graph of penalty based on the change in distance to the obstacle between past and current states. (**b**) A graph of advantage based on the change in distance to the obstacle between past and current states.

Remark 1: Assigning an intentional weight to the reward of $-550$ upon collision emphasizes the significance of the least desirable event (collision) commonly encountered in dynamic environments. This weighting aims to instill a recognition of the risk associated with collision states during the learning process. Additionally, in designing functions related to the destination, a fixed penalty is used. This is intended to continuously impose penalties of a magnitude similar to the maximum positive reward $+8.8$, aiding in policy formulation for reaching the destination. In the process of designing rewards related to obstacles, we use the exponential functions for both the advantage and penalty in a similar form. This aims to introduce a step-wise perception of the risk associated with collisions.

Additionally, the use of a large-scale function is employed due to the limited conditions and time in which the function operates, seeking to exert a robust influence during operation.

### 3.4. Constituent Networks of the DDPG

The DDPG consists of an actor network, which approximates the policy, and a critic network, which evaluates the value of the policy. Both networks are based on a multilayer perceptron (MLP) structure comprising fully connected layers. To ensure learning stability, target networks are also constructed for each network. Figure 6 illustrates the structure of the actor network. This network uses state $s_t$ as an input, which passes through two hidden layers, each consisting of 500 nodes, to generate two values representing linear and angular velocities.

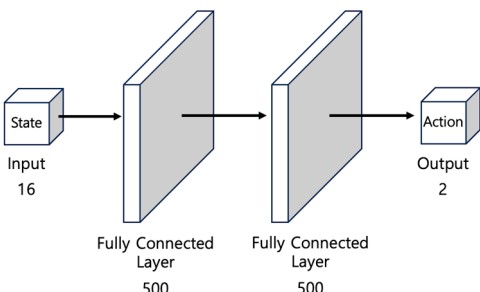

**Figure 6.** The structure of the MLP-based actor network for determining the optimal actions. It takes 16 states as input and outputs 2 actions.

Figure 7 illustrates the critic network. The input consists of two components: state $s_t$ and action $a_t$. After passing through the hidden layers, each containing 250 nodes, the intermediate output is incorporated into the second hidden layer with 500 nodes. Finally, the network generates a single value as the output, namely the Q-value for the given state and action. This value is used to evaluate the policy.

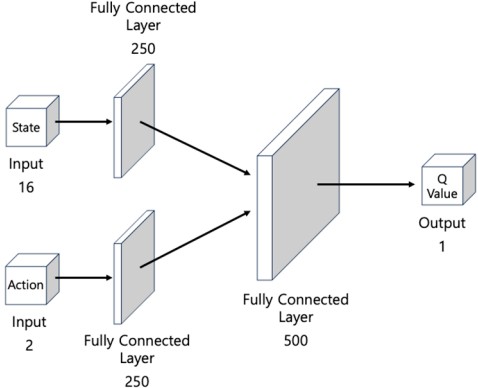

**Figure 7.** The structure of the MLP-based critic network for evaluating the determined actions. It takes 16 states and 2 actions input and outputs 1 Q-value.

The actor network is trained to maximize the Q-value, i.e., the output of the critic network. The network parameters are updated using the gradient ascent method with the loss function $L_i^a$ defined as follows:

$$L_i^a = -\sum_i Q(s_i, \mu_{\theta\mu}(s_i); \theta_Q), \tag{16}$$

where $\mu_{\theta\mu}$ denotes the deterministic policy and $\theta_Q$, $\theta_\mu$ denote the weights of the critic and actor networks, respectively. Although the critic network also updates its parameters

using the gradient descent method, the loss function is defined as a smooth L1 loss using the Q-value and labeled as follows:

$$l_i = Q(s_i, a_i; \theta_Q) - y_i \tag{17}$$

$$L_i^c = \begin{cases} \frac{1}{n}\sum_i 0.5l_i^2 & \text{if } |l_i| < 1 \\ \frac{1}{n}\sum_i |l_i| - 0.5 & \text{otherwise} \end{cases}' \tag{18}$$

where $y_i = r_i + \gamma Q\left(s_{i+1}, \mu_{\theta_\mu^-}(s_{i+1}); \theta_Q^-\right)$, $\gamma$ is the discounting factor, and $\theta_\mu^-$ and $\theta_Q^-$ denote the weights of the target actor and target critic networks, respectively.

*3.5. Generating Alternate Data with HER*

In reinforcement learning, the common approach to collecting training data is to store transitions $(s_t, a_t, r_{t+1}, s_{t+1})$ in a memory buffer. These transitions are generated after the agent performs an action. The learning process is initiated after a certain number of data transitions are accumulated in the memory buffer. In this process, finding the optimal policy is challenging due to the low probability of achieving the goal through exploration. This difficulty is exacerbated, especially in environments with sparse rewards. To address this issue, we implement the HER by re-evaluating failed episodes to create successful trajectories. Algorithm 1 illustrates the detailed process of implementing the HER. G is a set of states to be re-evaluated as new destination states selected from failed trajectories. A failed episode occurs when the robot collides with walls or obstacles or experiences a timeout. In each failed case, HER is applied three times. In the case of a collision, the states corresponding to steps 5, 25, and 50 before the final state of the trajectory are designated as the new destinations. As shown in Figure 8, trajectories from the initial position to these new destinations are extracted. The white trajectory represents the original unsuccessful path, while the blue, green, and yellow trajectories signify new successful paths, each setting the state 5, 25, and 50 steps before as the updated destination. Rewards are then recalculated, contributing to the generation of successful experiences.

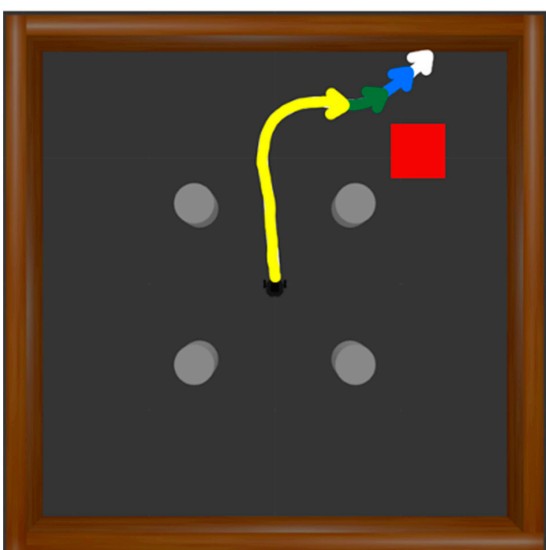

**Figure 8.** Generating successful trajectories based on HER in case of collision. The white trajectory represents the original failed path, while the blue, green, and yellow trajectories signify new paths aiming for the destination 5, 25, and 50 steps before, respectively.

Algorithm 1. The Hindsight experience replay algorithm applied in this study. Success and failure episodes are each applied three times, generating diverse paths for new successful experiences to enhance diversification.

---

**Algorithm 1. Hindsight Experience Replay**

---

1:    terminate time T
2:    after episode terminate,
3:    $G = \varnothing$
4:    if $s_T$ is collision
5:       $G = \{s_{T-5}, s_{T-25}, s_{T-50}\}$
6:    if $s_T$ is timeout
7:       $G = \{s_{T-50}, s_{T-150}, s_{T-250}\}$
8:    for $g' \in G$ do
9:       for $t = 0, T$ do
10:         $r' := R\left(s_t, a_t, p^{g'}\right)$
11:         if $r'$ is 550
12:            Break
13:         Store the transition $(s_t \parallel g', a_t, r', s_{t+1} \parallel g')$ » $\parallel$ denotes concatenation
14:      end for
15:   end for

---

When a timeout state occurs, the states 50, 150, and 250 steps before the terminal state are designated as the new destination, as illustrated in Figure 9. The white trajectory denotes the original failed path, while the blue, green, and yellow paths represent successful trajectories. Each trajectory sets the state 50, 150, and 250 steps before as the new destination, respectively, and the trajectories are extracted. Rewards are then recalculated, contributing to the generation of successful experiences.

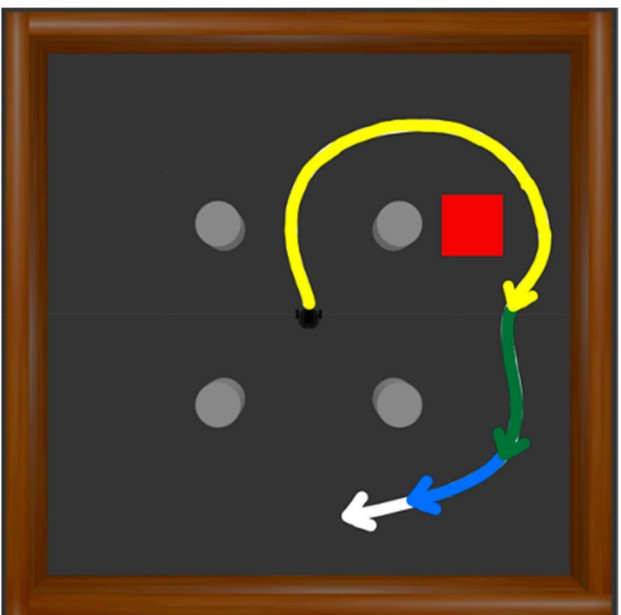

**Figure 9.** Generating successful trajectories based on HER in case of timeout. The white trajectory represents the original failed path, while the blue, green, and yellow trajectories signify new paths aiming for the destination 50, 150, and 250 steps before, respectively.

## 4. Experimental Results

### 4.1. Experimental Progress

To evaluate the effectiveness of the proposed method, we conducted identical experiments for each method by repeating the experiment 10 times.

(1) Case 1: Only the DDPG algorithm is used with the simplest reward system. This case serves as a baseline to evaluate the effects of the proposed technique.
(2) Case 2: Only the goal-based reward shaping method is applied to enhance the reward system.
(3) Case 3: Only the multifunctional reward shaping method is applied to enhance the reward system.
(4) Case 4: The HER technique is used.
(5) Proposed Method: Both multifunctional reward shaping and the HER technique are applied.

Given the nature of deep reinforcement learning, instances exist where discovering the optimal policy is not guaranteed, even within identical learning scenarios. This implies that under consistent configurations, the likelihood of identifying the optimal policy may fluctuate. To assess the robustness and reliability of the proposed methodology, the experiment was replicated ten times. To easily visualize the progress of policy optimization, the average rewards of the most recent 50 episodes were plotted and compared. To address the comparison challenge introduced by multifunctional reward shaping, the rewards obtained from all experiments were compared using the baseline reward system defined by Equation (13). In addition, following the completion of the learning phase, test-driving was performed using trained artificial neural networks. In this process, actions were determined without the addition of noise. Each driving test was conducted in both simulated and real environments. In detail, in Figure 10a, 100 episodes were conducted with the destination set randomly to the same settings as the learning environment in the simulation. In a real environment, as shown in Figure 10b, four fixed obstacles of different sizes are installed and one person acts as a dynamic obstacle that moves randomly. To test the adaptability of the optimal policy found by the proposed method, the actual environment is set up slightly differently from the learning environment, and the destination is set to each of the four corners of the space. We conducted a total of 20 test drives, with 5 tests for each corner.

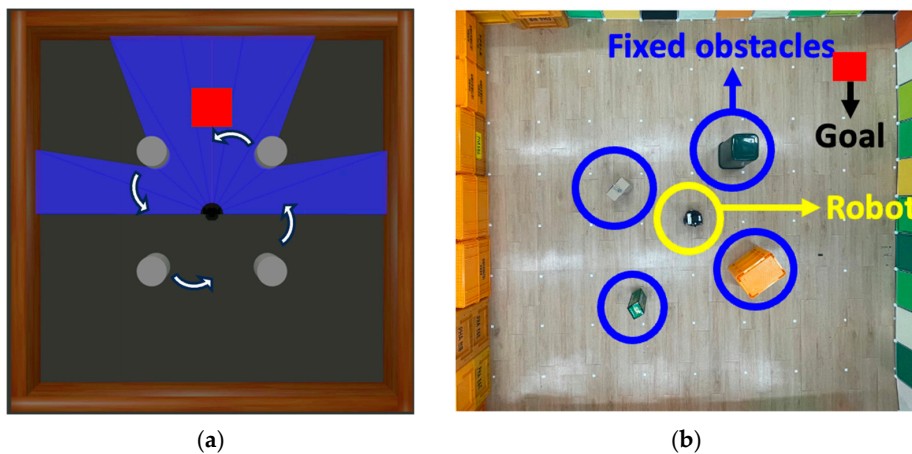

(**a**)                                              (**b**)

**Figure 10.** (**a**) The test-driving environment in simulation; (**b**) the real-world test-driving environment, distinct from the trained environment.

### 4.2. Experimental Results

Figure 11 is a graph depicting the case in which the best policy was found among 10 learning sessions conducted in each case. In case 1, learning was conducted with a sparse reward system based on the DDPG algorithm, and it converged to $-300$ points. Since the maximum number of actions is set to $+300$, we can confirm that it has converged to a sub-optimal policy that runs in place without colliding. In the remaining cases, converging to values greater than $-300$, it can be concluded that the policy successfully navigated to the destination while avoiding obstacles. However, analyzing the completeness of the policy, it can be observed that case 2 with goal-based reward shaping and case 3 with only

the HER technique have lower completeness compared to when the proposed method is applied. Case 4 with only the proposed reward-shaping technique and case 5 with both proposed reward shaping and the HER converged to relatively high values around +200. Therefore, it can be considered that policies were found to navigate through optimal paths, avoiding obstacles and reaching the destination, indicating higher completeness.

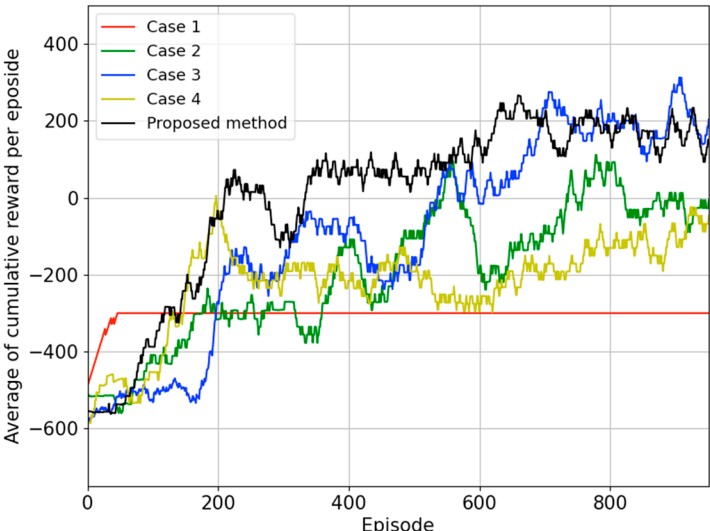

**Figure 11.** During the 10 trials for each case, the average reward curve during training for the model with the highest policy completeness.

Table 1 shows the training success rate and test-driving results for each case. First, the training success rate represents the ratio of trials that found the optimal policy out of 10 trials for each case. From the perspective of the training success rate, the case that applied both the proposed reward shaping and the HER simultaneously showed the highest success rate of 80%. In order to compare policies in terms of completeness, test-driving was conducted using the policy with the best performance for each case. The results of the simulated test-driving indicate that case 3, which applied only the proposed reward-shaping technique, had the highest success rate, and when combined with the HER technique, it reached 97%. When applying the proposed techniques, it was possible to find policies with relatively higher completeness than in other cases. In addition, it can be said that Case 3 and the proposed method have the best success rate even when driving in a real environment that is different from the learning environment, helping to find policies that can adapt to changes in the environment.

**Table 1.** Test-driving results for all cases in the simulation and the real world.

|  | Case 1 | Case 2 | Case 3 | Case 4 | Proposed Method |
|---|---|---|---|---|---|
| Training success | 0% | 40% | 50% | 30% | 80% |
| Test in simulation | 0% | 91% | 99% | 72% | 97% |
| Test in real-world | 0% | 90% | 95% | 5% | 95% |

Remark 2: Based on the experimental results, the proposed method is effective in determining the optimal policy for advanced autonomous driving in dynamic environments. The contribution of this study lies in demonstrating the potential for enhancing autonomous driving in dynamic environments by incorporating both destination and obstacle information using multifunctional reward-shaping techniques. Moreover, the proposed scheme implements both HER and multifunctional reward-shaping techniques, which have not been simultaneously deployed in previous studies. In particular, we implemented the reward-shaping technique to assist in achieving objectives and sub-objectives even during ongoing episodes and utilized the HER technique to balance data between failure and

success cases. Consequently, the proposed method successfully determined the optimal policy in most experimental cases.

## 5. Discussion

In the field of autonomous mobile robot navigation, the primary goal is to reach the destination while avoiding obstacles. The techniques based on SLAM have successfully implemented autonomous navigation in indoor environments by relying on pre-mapped surrounding information. However, it faces limitations when unexpected dynamic obstacles appear or when there are changes in the internal elements of the indoor environment, necessitating map reconstruction. Research is underway to apply reinforcement learning to the autonomous driving of mobile robots. In the other study, reward shaping was applied with the DDPG, but there was a need for improvement in adaptability to dynamic obstacles. To address this limitation, this study proposes a technique that simultaneously applies the HER and multifunctional reward shaping. The objective is to achieve autonomous driving by effectively handling dynamic obstacles. Verification through test-driving in both simulation and real-world environments demonstrates the effectiveness of our approach. The HER proves valuable by generating successful experiences from failed ones, addressing the imbalance in experience data, and aiding in finding optimal policies. The multifunctional reward shaping continuously provides information about the goal and obstacles, facilitating in finding policies that avoid obstacles while reaching the destination. The training success rate of our proposed technique reached 80%, showcasing its effectiveness. From the perspective of overall driving success, our method achieved a success rate of over 95% in both simulation and real-world test driving, validating its effectiveness. Notably, despite differences in the composition of the training and real-world environments, the 95% navigation success rate achieved highlights the adaptability of the reinforcement learning-based autonomous driving technique to environmental changes.

Compared to SLAM techniques, our proposed approach exhibits advantages in environmental adaptability. This study demonstrates that intuitive ideas, such as those presented in our technique, can enhance performance and offer advantages in terms of implementation complexity. This underscores the adaptability of reinforcement learning-based autonomous driving technology to dynamic environmental changes.

## 6. Conclusions

We propose a technique that adopts the concepts of both multifunctional reward-shaping and HER to implement the autonomous driving of a mobile robot based on reinforcement learning in a dynamic environment. Reward shaping is used to design a reward system that induces actions to reach a destination while avoiding obstacles. The specific reward system was constructed by designing functions that provide information about the destination and obstacles, respectively. In addition, to balance the experiences of failure and success, we implemented the HER, which generates success experiences from failure experiences. Therefore, the proposed method addresses the sparse reward problem and aids in finding the adaptive optimal policy in dynamic environments. The proposed approach, combining the reward shaping and HER techniques, was validated through simulation and real-world test-driving, demonstrating its effectiveness in finding optimal policies. In particular, the proposed method demonstrated effectiveness in finding adaptable optimal policies, as evidenced by the high success rate in real-world environments different from the training setting.

**Author Contributions:** Conceptualization, M.P. and N.K.K.; methodology, M.P.; software, M.P.; validation, M.P. and N.K.K.; writing—original draft preparation, M.P.; writing—review and editing, C.P. and N.K.K.; visualization, M.P.; supervision, C.P. and N.K.K.; project administration, N.K.K. All authors have read and agreed to the published version of the manuscript.

**Funding:** This research was supported by the Korean Federation of Science and Technology Societies Grant and the Korea Institute for Advanced Technology (KIAT) grant funded by the Korean Government (MOTID (P0008473, HRD Program for Industrial Innovation) and by the National Research Foundation of Korea (NRF) grant funded by the Korean government (MSIT) (NO.RS-2023-00219725).

**Institutional Review Board Statement:** Not applicable.

**Informed Consent Statement:** Not applicable.

**Data Availability Statement:** Data are contained within the article.

**Acknowledgments:** The authors express their sincere appreciation for all those who contributed to this study.

**Conflicts of Interest:** The authors declare no conflicts of interest.

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
