# Peer review of "Autonomous Driving of Mobile Robots in Dynamic Environments Based on Deep Deterministic Policy Gradient: Reward Shaping and Hindsight Experience Replay"

_biomimetics, doi:10.3390/biomimetics9010051_

Round 1
Reviewer 1 Report
Comments and Suggestions for Authors
Please, see the attached PDF.

Comments on the Quality of English LanguageQuality of English is fine and no relevant issues have been detected.
Reviewer 2 Report
Comments and Suggestions for Authors
Summary:
The paper proposes a reinforcement learning-based method for autonomous driving of a mobile robot in a dynamic environment with obstacles, using two additional techniques: multifunctional reward shaping and hindsight experience replay. The mobile robot successfully reaches its destination without colliding with obstacles by utilizing these techniques.
The primary limitation of this work lies in the performance evaluation section. In my opinion, there is a noticeable shortfall in effectively demonstrating the potential advantages of the proposed methods within the given scenario.
Comments:
1. I recommend that the authors enhance the structure of their paper by adding a section on related work. This section should provide a thorough analysis of previous methods in Autonomous Driving of Mobile Robots and elucidate how the proposed methods improve upon existing ones.
2. The fonts and lines in Figures 9 to 12 need to be enlarged to enhance readability. Currently, it is challenging to comprehend the details in these diagrams.
3. The authors have not convincingly demonstrated the effectiveness of the Multifunctional reward shaping in their experiments. It would be beneficial if the impact of equations 13, 14, and 15 were illustrated in a separate diagram to clearly convey their effects.
4. Can you elaborate on the role of hindsight experience replay in your method? How does it contribute to overcoming the challenges of sparse rewards and data imbalance between success and failure cases in training the autonomous driving system?
5. In practical scenarios, mobile robots often encounter unpredictable elements. How does your method ensure adaptability and robustness in such unpredictable environments?
6. Authors do not show How the proposed method address the computational efficiency and real-time processing requirements essential for autonomous mobile robotics in dynamic environments.
7. Are there any limitations or potential drawbacks of your method that should be considered, especially when thinking about scaling or applying it to different types of mobile robotic systems or environments?
8. Can you explain why case 2 and case 3 outperform proposed method in trail 6 and trail 9. Are there specific scenarios or conditions where this combined approach is particularly advantageous?
9. Regarding the experimental setup with the Turtlebot3 and moving obstacles in a gazebo simulation, what were the primary metrics used to evaluate the effectiveness of your method? How do these metrics correlate with real-world performance in dynamic environments?
10. Could you discuss any limitations or challenges you encountered while integrating these techniques into a single autonomous driving system? How might these affect the scalability or adaptability of your method to different types of environments or robotic platforms?
Comments on the Quality of English Language
proofread carefully
Round 2
Reviewer 1 Report
Comments and Suggestions for Authors
No further comments. The revision improved the clarity and completeness of the manuscript by addressing all Round 1 comments.
Comments on the Quality of English LanguageNo relevant issues were found.
Author Response
Thank you for your comments.